# Ameliorative Effects of Luteolin and Activated Charcoal on Growth Performance, Immunity Function, and Antioxidant Capacity in Broiler Chickens Exposed to Deoxynivalenol

**DOI:** 10.3390/toxins15080478

**Published:** 2023-07-26

**Authors:** Mubashar Hassan, Yanan Wang, Shahid Ali Rajput, Aftab Shaukat, Ping Yang, Muhammad Zahid Farooq, Qianhui Cheng, Mehboob Ali, Xiaomei Mi, Yu An, Desheng Qi

**Affiliations:** 1Department of Animal Nutrition and Feed Science, College of Animal Science and Technology, Huazhong Agricultural University, Wuhan 430070, China; mubashar.hassan@webmail.hzau.edu.cn (M.H.); ynwang@webmail.hzau.edu.cn (Y.W.); cqh@mail.hzau.edu.cn (Q.C.); mixiaomei@webmail.hzau.edu.cn (X.M.); ay2000@webmail.hzau.edu.cn (Y.A.); 2Department of Animal Feed and Production, Faculty of Veterinary and Animal Sciences, Muhammad Nawaz Shareef University of Agriculture, Multan 66000, Pakistan; dr.shahidali@hotmail.com; 3College of Veterinary Medicine, South China Agricultural University, Guangzhou 540642, China; aftabshaukat40@gmail.com; 4State Key Laboratory of Agricultural Microbiology, Hubei Hongshan Laboratory, Frontiers Science Center for Animal Breeding and Sustainable Production, College of Animal Science and Technology, Huazhong Agricultural University, Wuhan 430070, China; zahid.farooq@uvas.edu.pk; 5Department of Animal Sciences, University of Veterinary and Animal Sciences (Jhang Campus), Lahore 54000, Pakistan; 6State Key Laboratory of Agricultural Microbiology, College of Veterinary Medicine, Huazhong Agricultural University, Wuhan 430070, China; mehboobali3444@gmail.com

**Keywords:** deoxynivalenol, luteolin, activated charcoal, growth performance, antioxidant capacity, immunity, broilers

## Abstract

Deoxynivalenol (DON, Vomitoxin) is a threatening mycotoxin that mainly produces oxidative stress and leads to hepatotoxicity in poultry. Antioxidant dietary supplements dramatically boost immunity, safeguarding animals from DON poisoning. Luteolin (LUT) is an active plant-derived compound that poses influential antioxidants. This study explored the effectiveness of LUT in combination with activated charcoal (AC) in detoxifying DON in broilers. The 180 one-day broiler chickens were allocated into five different groups having six replicates in each group, provided with ad libitum feed during the trial period (28 days) as follows: in the control group, basal diet (feed with no supplementation of LUT, AC or DON); in group 2, a basal diet added with 10 mg/kg DON from contaminated culture (DON); in group 3, a basal diet augmented by 350 mg/kg LUT and DON 10 mg/kg (DON + LUT); in group 4, a basal diet supplemented by DON 10 mg/kg + AC 200 mg/kg (DON + AC); and in group 5, a basal diet supplemented by 10 mg/kg DON + 350 mg/kg LUT + 200 mg/kg AC (DON + LUT + AC). Concerning the control group, the DON-treated broilers demonstrated a significant decrease in growth performance (*p* < 0.05) and serum immunoglobulin (*p* < 0.05) contents, negatively changing the serum biochemical contents and enzymatic activities and an increase in histopathological liver lesions. Furthermore, DON substantially increased (*p* < 0.05) malondialdehyde (MDA) concentration and decreased total superoxide dismutase (T-SOD), catalase (CAT), and glutathione peroxidase (GSH-Px) levels in the serum and liver. The intake of AC and LUT to the DON-contaminated diet decreased DON residue in the liver and potentially reduced the adverse effects of DON. Considering the results, supplementation of LUT with mycotoxin adsorbent has protective effects against mycotoxicosis caused by DON. It could be helpful for the development of novel treatments to combat liver diseases in poultry birds. Our findings may provide important information for applying LUT and AC in poultry production.

## 1. Introduction

Poultry meat is a unique source for the biological processing of richly supplied animal protein [1]. It constitutes about 102.9 million tons of meat production [2]. The world’s highly ranked meat-producing countries are China, the US, Brazil, and Russia [1]. Broiler meat is found to be one of the cheapest sources richly supplied with protein in different forms of foods [1]. The consumption of poultry meat is predicted to increase to 152 Mt worldwide, estimating an increase of up to 52% of the total meat consumed [3]. The prognosticated growth rates in the consumption of poultry per capita contemplate the substantial role in the national foods of different populations in developing countries, especially in China and India, from 2005 to 2030 [4]. The production will increase to 3.6% and 3.5% annually, respectively, as India, Brazil, and China continuously expand their markets. However, the poultry industry still faces several bottlenecks, such as mycotoxins, limiting further applications [5]. Mycotoxins are present in about 25% of total supplies of world feed [6].

Moreover, DON is the most toxic Fusarium mycotoxin in animal feeds [7]. Most cereals crops, such as barley, oats, rye, and maize, are strongly affected by the action of DON. Recently, scientists have found that DON, even at low concentrations allowed by the E.U., caused mycotoxicosis and harmfully affected the gut and performance of broilers [8]. In contrast, mycotoxins impact on broiler performance will effectively rely on feed contamination [9]. Additionally, mycotoxins are potentially absorbed in gut tissues, and the intestinal cells are quickly imposed harmful effects, which damage the proper and normal gut function [8]. The mitochondrial transport chain actively releases reactive oxygen species (ROS) into muscle cells due to broiler metabolic rates and rapid growth. Broilers are also affected by mycotoxins, metabolites of mycotoxins, and microbes found in animal feed that increase free radical production and inflammation [10]. Some valuable strategies to mitigate the harmful effect of mycotoxins are based on the degradation and absorption of mycotoxins in feed [11]. In addition, Fusarium mycotoxins have been shown to negatively impact the broiler antioxidant defense system in the liver [12]. In the presence of ROS, mitochondrial oxidative stress and lipid oxidation is caused, and the lipid oxidation [13] is associated with increased malondialdehyde (MDA) formation, an increase in lipid peroxidase activity, and reduced activities of superoxide dismutase (SOD) and glutathione (GSH).

The flavonoid LUT is a plant derivative in parsley, chamomile tea, celery, fruits, perilla leaf, and green pepper. It possesses antioxidant, anti-inflammatory, absorbing oxygen, anti-cancer, and DNA repair properties [14,15]. In prior experimental trials, LUT has a broadly positive effect as cardioprotective and neuroprotective [16]. Furthermore, LUT greatly shields LPS-induced liver damage and oxidative stress in feeding mice [17]. Additionally, LUT induces body weight gain and improves immune injury in DON-induced mice [18]. Activated charcoal can efficiently absorb mycotoxins [19], reverse the suppression of mycotoxin-induced immune response [20], and rapidly improve the growth and performance of liver functioning of broilers [21]. This study aims to assess the toxic effects of DON and the protective effects of LUT and AC on growth performance, biochemical parameters, liver histopathology, and DON residues exposed to DON-contaminated feed in broilers.

## 2. Results

### 2.1. Compared with the DON Group, LUT and AC Supplementation Significantly Improved the Growth Performance

Growth performance is summarized according to the five treatments in Table 1. It was found that the DON-contaminated group noted the average daily gain and daily feed intake lowest during the entire experiment period (*p* < 0.05). ADG and ADFI increased when LUT was added to DON-containing diets (*p* < 0.05). Dietary DON negatively impacted the feed conversion ratio (FCR) of broilers (*p* < 0.05). There was a significant improvement in FCR with DON + LUT and DON + AC groups compared to the DON group (*p* < 0.05). Additionally, the LUT and AC treatment did not negatively affect broiler performance. The results demonstrated that LUT and AC eliminated DON toxic effects on growth.

### 2.2. Serum Biochemical Analysis Based on Biochemical Criteria

The serum biochemical changes were also investigated after adding LUT and AC to the DON-contaminated diet. According to Table 2, DON-contaminated feed affected the serum biochemical profile in a negative way (*p* < 0.05) compared to other groups. It has been shown that supplementation with LUT and AC alleviated this toxic effect of DON. Serum biochemical indicators such as ALT, AST, TG, and ALP were enhanced by approximately 63%, 34%, 47%, and 39%, respectively, in the DON-fed group.

However, with LUT supplementation, ALT, AST, TG, and ALP significantly decreased by 41.05%, 34.64%, 40.22%, and 29.07%, respectively. There were no significant differences in adding LUTs to the feed compared to the control group. The level of serum biomarkers was also decreased with the addition of both LUT and AC. While globulin, albumin, and total protein (TP) significantly increased, compared to the diet alone contaminated with DON, they increased less than in the LUT and DON groups.

### 2.3. Immunoglobulin Levels in Serum

According to Figure 1, broilers fed a diet containing DON had altered immune responses compared with the control group; DON significantly reduced IgM, IgA, and IgG by 46.15%, 45.07%, and 38.30%, respectively. As a result of LUT treatment, the DON toxic effect on serum immunoglobulin was alleviated (*p* < 0.05) in contaminated diets. However, when supplemented with a contaminated diet, there was no significant difference between the diets of LUT and AC. The results indicated that DON damaged the immune system. Adding LUT and AC to a diet contaminated with DON was able to counteract its immunosuppressive effects.

### 2.4. Analyses of Serum Antioxidants

The effect of LUT and AC on serum antioxidant indices in broilers treated with DON is summarized in (Table 3). Broilers exposed to DON in their food had greater serum MDA levels than the control group. A significant reduction in MDA levels was observed when LUT was added to diets contaminated with DON, with less difference between the two groups (DON + AC and DON + LUT + AC). DON administration reduced antioxidant enzyme activity, as measured by T-SOD, GSH-Px, and CAT, compared to the control group. In broiler chicken feed, LUT significantly increased antioxidant enzyme activities. The satisfied results were also found by adding LUT and AC to a DON-adulteration feed.

### 2.5. LUT and AC Prevent Damage to the Liver Caused by DON in Broilers

Hepatic histological changes are illustrated in (Figure 2). In the control (Figure 2A) and DON + LUT (Figure 2C) groups, respectively, there were no histopathological changes in the broiler’s livers, while little change in the DON + LUT + AC group (Figure 2E) was observed. According to the results of the histological analysis, broilers who consumed DON alone suffered significant liver damage (Figure 2B). Comparatively to LUT and AC diet-fed birds (Figure 2B), the liver tissue of the DON group had hyperplasia of the bile ducts, diffused watery degeneration, and hepatocyte necrosis. Broiler hepatic parenchyma was protected from injury by adding LUT and AC to DON diets (Figure 2C–E).

Liver weight was shown to be reduced in birds when they were fed DON-affected feed (*p* < 0.05). The DON-fed group gained significantly more relative liver weight (Figure 3). Additionally, histological findings demonstrated that LUT alleviated DON-induced toxicity in broilers.

### 2.6. LUT and AC Improve Liver Antioxidant Parameters

The presence of DON in broiler diets adversely affected (*p* < 0.05) the liver antioxidant status (Table 4). There is a clear difference between the two groups, with an increase in MDA by 54% (*p* < 0.05) equated to the control group. When LUT and AC were supplemented with exposure to a contaminated diet, the MDA content was reduced by 39.57% (*p* < 0.05) compared to the DON group. The antioxidant activities of the DON + AC and DON + LUT + AC groups were significantly improved by 42.04% and 40.98%, respectively, in comparison with the DON group. In addition, treatment with LUT and AC along with the DON diet significantly stimulated the liver’s antioxidant system to counteract the oxidative damage caused by DON.

### 2.7. Residues of DON in the Liver

According to (Figure 4), the liver of broilers fed DON-contaminated diets showed DON residues. Broilers fed DON + LUT exhibit fewer DON residues in their livers. Broilers fed DON-contaminated diets had detectable levels of DON (0.29 µg/kg) in their livers. A significant reduction of 37.93% and 13.79% in DON residues in the liver was observed when DON + AC and DON + LUT + AC were added to the diet matched to the DON alone group.

## 3. Discussion

Mycotoxins in cereals and animal feed represent severe health and financial dangers to people and animals. It is challenging to acquire feed utterly free of DON contamination because of the widespread presence of this mycotoxin in contaminated grains [8]. DON contamination is a major issue in the food and feed industries because of its wide range of harmful consequences, including hepatotoxicity, immunotoxicity, genotoxicity, and carcinogenicity [8]. It has been shown in a study by [8] that 15 mg/kg DON reduced broiler chicken body weight gain after 42 days and altered feed conversion ratios. It is clear from this study that broiler chickens eating DON have toxic effects on their growth performance. Consistent with prior studies, feeding broiler feed polluted with 15 mg/kg DON considerably declined ADG and ADFI, negatively affecting broiler chickens’ cumulative feed gain ratio [22]. Evidence supports these adverse effects of DON due to its association with reluctance, anorexia, impaired protein synthesis, immune system impairment, and lipogenesis [23]. The heavy growth of fungi in the feed probably contributed to all these changes. Fungal growth and mycotoxin production primarily depended on carbohydrates [22]. Some studies have found that when mycotoxins are present in feed at concentrations close to the maximum permitted limit, weight gain and feed intake rates drop [23]. In younger broilers, mycotoxins affected growth more adversely than in older ones [24], while older broilers had significantly reduced feed intake and growth rates [25]. The authors claim that because the older birds absorbed more mycotoxins (more feed), they had higher FCRs and opposite consequences.

The growth performance was slowed when tainted feed was given to Ross broilers [25]. These findings corroborate earlier research showing a marked reduction in ADG and ADFI in broilers fed diets polluted with DON. DON negatively impacted the broiler’s cumulative feed gain ratio. However, these parameters were improved by adsorbents, particularly by the mixture of bentonite with kaolin and *Saccharomyces cerevisiae* and bentonite [26]. Previous research has shown that kaolin and bentonite can reduce mycotoxin levels in feed contaminated with DON and aflatoxin. Even though neither adsorbent has been shown to reduce mycotoxins in chicken feed effectively, it is widely utilized because of its low primary toxicity. The digestibility of the nutrients was increased by adding adsorbents to the meal [26]. Mycotoxin binder’s primary purpose is to adsorb mycotoxins and stop them from being absorbed by the digestive system [26]. Mycotoxin-contaminated poultry feed may benefit from adsorbent combinations because these adsorbents appear to bind to only a limited group of mycotoxins without showing an affinity for other mycotoxins [27]. Intestinal mesophilic aerobic bacteria were significantly inhibited by combining kaolinites and activated charcoal with zearalenone (ZEN) and DON, administered individually and simultaneously [28].

Furthermore, LUT can mitigate the harmful effects of DON on growth performance. LUT was said to considerably increase the body weight of mice suffering from aflatoxin B1 (AFB1) [29]. In contrast to the DON group, we discovered that adding LUT and AC to meals contaminated with DON dramatically improved ADFI, ADG, and FCR (Table 1). As a result, the effectiveness of LUT and the adsorbent combinations varied in lowering mycotoxin levels in broiler feed.

Liver function and pathology can be learned from serum ALT, AST, and ALP measurements. ALT predominates in the cytoplasm, whereas AST predominates in the mitochondria [30]. It is well known that DON inhibits the synthesis of proteins [31]. We observed increased serum levels of AST and ALT after DON exposure, suggesting damage to the liver cytoplasm and mitochondria. The DON group also reported considerably greater serum ALP levels than the control group, indicating that liver damage had occurred. Total protein, albumin, and globulin contents were all found to be significantly lower.

Unlike serum, biochemical values of AST, ALT, TG, and ALP were significantly increased after providing DON-polluted feed, consistent with other research and this study’s findings [32]. It has been shown that DON causes hematological toxicity, liver dysfunction, and steatosis, and LUT can alleviate these symptoms. Nevertheless, the DON + LUT group showed no effect, likely due to LUT enhancing immunity and protecting liver function. A DON-contaminated feed with LUT and AC increased TP, ALB, and globulin and decreased serum indices (AST, ALT, TG, and ALP) in broilers (Table 2). As a result of these findings, LUT and AC significantly reduced the toxic effects of DON on the liver. Immunoglobulin levels were significantly lower in broilers given diets polluted with DON. The previous study found that DON dramatically lowered serum immunoglobulins (IgA, IgG, and IgM) and increased haptoglobin levels in mice fed DON [33]. Inhibitors of protein synthesis, such as DON and T-2 toxins, stimulate the production of interleukin 1 and interleukin 2 [34]. In addition to intestinal antigen exposure, the oral challenge suppresses specific dietary protein-induced IgG and IgM responses. While it is widely accepted that IgA is the primary immunoglobulin found in mucosal secretions, it is not entirely understood what IgA does in the serum [35]. According to our results, serum IgA, IgG, and IgM concentrations significantly increased with LUT and AC when combined with contaminated diets containing DON (Figure 1). A decrease in immunity was observed as a result of DON. Antioxidants and ROS maintain a relative balance within the intracellular antioxidant defense system.

In contrast, oxidative stress occurs when antioxidants and ROS are out of balance [36,37]. The amount of ROS produced by DON increases, which attacks the lipids in the cell membranes, thus altering their fluidity and permeability and causing oxidative damage to the cells. Oxidative stress and mitochondrion function are often detected through GSH and MDA concentrations [38,39]. Additionally, SOD is an important antioxidant [40,41]. The induction of oxidative stress and lipid peroxidation is a well-established impact of DON. Changes in fatty acid content make lipids more reactive to oxygen-free radicals; the glutathione redox system counteracts this effect. DON caused a drop in GSH-Px activity and other indicators of the glutathione redox system relative to baseline. Researchers found that adding LUT to quail livers alleviated oxidative stress and mitochondrion injury caused by Imidacloprid [14]. The liver and serum antioxidant levels were significantly reduced when LUT and AC were added to the feed of broilers exposed to DON (Table 3). As a result, LUT and AC are crucial in reducing oxidative damage caused by DON in broiler chickens.

Mycotoxins produced a variety of impacts on chickens, including liver disease and changes in relative organ weights [41]. Histological findings demonstrated that LUT protected against DON-induced damage. For the first time, it appears that LUT can alleviate liver damage caused by DON in broilers. Surprisingly, dietary LUT supplementation reduced the histopathological changes induced by DON. These findings were consistent with a prior investigation that found LUT protective against AFB1-induced hepatic damage in rats [29]. In addition, the DON group had a marked rise in liver weight compared to the control group. Previous studies on the harmful effects of DON on liver weight relative to body weight have found similar results. Hepatomegaly, caused by fat buildup in the liver, is a side effect of DON [42,43]. We found that the dramatic rise in liver relative weight seen in the DON group could be mitigated by adding LUT and AC to the contaminated meal (Figure 3). These results showed that LUT and AC prevent liver damage caused by DON. Liver detoxification is mainly accomplished through metabolic conversion and biliary excretion of wastes and xenobiotics [44]. Several studies have demonstrated that DON intake is associated with high levels of liver damage in piglets [45].

In the present study, we observed the level (0.29 µg/kg) of DON residues in the liver of broilers fed on the DON-contaminated diet. A previous study reported that piglets fed contaminated moldy corn for 29 days had higher levels of DON residue in their livers [46]. Another study showed that duck livers with a diet containing 3 to 6 mg/kg DON showed hemorrhages and edema in the gall bladder [47]. In addition, rats fed a DON diet for five days and chickens given 2.5 mg/kg of DON were found to have residues of the DON in their livers [46,48,49]. Residue levels may differ due to different animal and diet types, concentrations of DON, durations, and tolerances of DON. Compared with the DON group, LUT and AC have significantly reduced the levels of DON residues in the liver in the current study. There is a possibility that LUT and AC protect against DON because they biotransform DON in the gastrointestinal tract, reducing its absorption, which in turn reduces DON residues in the liver.

## 4. Conclusions

To sum up, as a result of improving growth performance, antioxidant capacity, immunity, serum biochemical profiles, and histopathological lesions, LUT and AC lowered the inflammatory response of DON in broilers, and LUT decreased liver DON residues. There is a further need to explore the beneficial effects of LUT and AC in other species and their mechanism of action, and we suggest that LUT and AC could be used as promising poultry feed additives to counteract DON toxicity.

## 5. Materials and Methods

### 5.1. Analysis and Preparation for the DON Toxin

The College of Plant Science and Technology, Huazhong Agricultural University, China, donated the *Fusarium graminearum* strain W3008. As described in previous research, potato dextrose agar was used to produce mature spores [50]. Briefly, A 1 L conical flask was filled with 300 g ground maize and 48 g rice with 150 mL sterilized distilled water and autoclaved for 15 min. Each flask was injected with 1 × 10^6^ spores/gram of *F. graminearum* and incubated for 28 days at 30 °C and 80% relative humidity. The flasks were shaken weekly to ensure proper growth of the fungus. The mold-contaminated samples in each flask were mixed, dried overnight at 65 °C in an air oven, and then tested for DON content. Approximately 350 mg/kg DON was present in the resulting moldy product. The DON content was assessed using Agilent High-Performance Liquid Chromatography (HPLC) 1260 series (Waldbronn, Germany) and an AgraQuant^®^ DON ELISA Test Kit following the manufacturer’s instructions [51,52,53,54,55,56].

### 5.2. Animals and Experimental Design

The LUT (#491-70-3, purity ≥ 98%) and AC (#7440-44-0) were provided by Shanghai Aladdin Biochemical Technology Co., Ltd., Shanghai, China. A commercial hatchery provided an aggregate of 180-day-old broiler chicks (Hunan Liyang Ecological Agriculture Co., Ltd., Changsha, China). After letting the birds rest for four days, they were randomly split up into five groups of six replicates in each group (n = 36 birds per treatment) based on the following five feeding trials: (1) BD, which did not contain either LUT, AC, or DON (Control); (2) BD augmented by 10 mg/kg DON from culture (DON); (3) BD supplemented with 350 mg/kg LUT and DON (LUT + DON); (4) BD supplemented with DON + AC 200 mg/kg (AC + DON); and (5) BD supplemented with DON + LUT + AC (DON + LUT + AC). For four weeks, the chicks were raised in chicken battery cages (six chicks in each battery cage), which consist of four stainless steel stages under normal temperature and hygienic conditions.

The humidity level was controlled between 60% and 75%. Throughout the trial, the birds’ health was continuously checked. The experiment was carried out in an environmentally friendly manner. Throughout the study period, tube feeders and nipple drinkers were used to administer ad libitum diets and water throughout the trial period. Basal diet formulation is shown in (Table 5). As mentioned above, mold harboring DON was used to contaminate the diets artificially, and the mold was spread throughout the feed for maximum effectiveness. National research specified the basal diet for starting chicks and included it in the feed (Table 5).

### 5.3. Sample Collection and Measurement

Each replicate of the trial was weighed weekly, and feed consumption was measured each week (four weeks). The following variables were calculated: BW, ADFI, ADG, and FCR. Each replication selected one bird near average weight at 28 days. Blood samples were taken by puncturing the bird’s wing vein after a 12 h fast. A centrifuge machine (5804R, Hamburg, Germany) was used to centrifuge blood samples. Biochemical, immunoglobulin, and serum antioxidant contents were determined from serum. The birds were sacrificed after taking blood samples, and their livers were quickly taken out and weighed. The remaining portion of the liver of the broilers was stored at −80 °C to conduct further experiments.

### 5.4. Histopathological and Biochemical Analysis of Serum

Serum samples were tested for the following parameters: TP, ALB, globulin, AST, ALT, TG, and ALP. The serum samples were analyzed using an automatic biochemistry analyzer (Beckman Synchron CX PRO, Fullerton, CA, USA). An embedding process of 10% neutral buffered formalin was followed by cutting liver tissue into 5 mm thick slices, which were then stained with hematoxylin and eosin (H & E). Microscopically, liver sections from all broilers were observed.

### 5.5. Detection of Antioxidative Enzymes in Serum and Liver

The liver tissue (0.1 g) was chopped into small pieces and homogenized in cold physiological saline (0.9 mL) to ascertain the quality of the tissue (Ningbo, China). The homogenate was centrifuged for 10 min at 4 °C at 12,000× *g*. After that, we collected the supernatant and put it in the freezer at −80 degrees Celsius until further investigation. A spectrophotometric analysis was performed to conclude the activities of antioxidant enzymes (CAT, GSH-Px, and T-SOD), as well as the concentration of MDA in serum and hepatic supernatants by the Jiancheng Bioengineering (Nanjing, China), as well as the specifics of the techniques used to test the commercial kits according to their protocols.

### 5.6. Analyses of Serum Immunoglobulin Concentrations

Serum, IgM, IgA, and IgG levels were determined with commercially available kits (Nanjing Jiancheng Bioengineering Institute, Nanjing, China). The measurements were carried out according to the instructions provided with the detection kit.

### 5.7. HPLC Analysis of DON Residues in Liver

Liver samples were frozen at −20 °C and verified for DON residues. A total of fifteen birds were used in this analysis (three birds from each treatment were selected to check DON residues in the liver). According to [57], DON residues in the liver were evaluated using a slightly modified approach. Thawed liver samples (20 g) were homogenized and mixed for 30 min with NaCl (2.5 g) in a mixture. An Agilent HPLC 1260 series (Waldbronn, Germany) equipped with a C_18_ column was then used to analyze the samples. The solvent mixture of 60:30:10 water, methanol, and acetonitrile was utilized at a 1 mL/min flow rate.

### 5.8. Statistical Analysis

One-way ANOVA was employed to examine the statistical significance between means. Duncan’s test was used when more than two differences were being compared. All data are provided as the standard error of means, with the significance threshold set at (*p* < 0.05) for all measurements. IBM SPSS Statistic 22 was used for the analysis (Armonk, NY, USA).

## Figures and Tables

**Figure 1 toxins-15-00478-f001:**
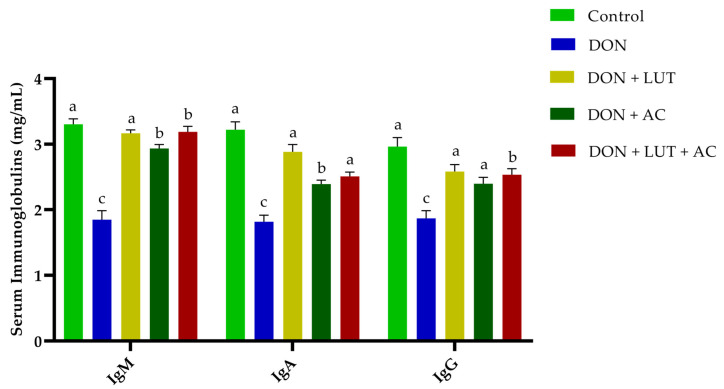
The effect of LUT and AC on DON-induced damage in serum immunoglobulins of the broilers (*p* < 0.05). Significant differences were labeled with different letters a–c. DON: deoxynivalenol, LUT: luteolin, AC: activated charcoal.

**Figure 2 toxins-15-00478-f002:**
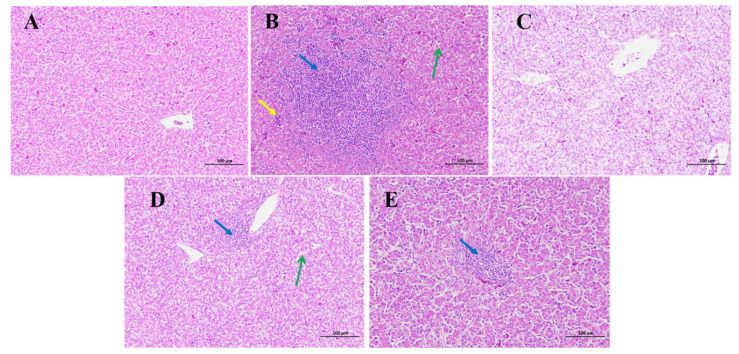
The effect of LUT and AC on DON-induced liver histopathology of broilers. Hematoxylin and eosin (H & E) analysis of the liver tissue of the broilers. Magnification × 200; Scale bar: 100 µm. (**A**) Control; (**B**) DON; (**C**) DON + LUT (**D**) DON + AC; (**E**) DON + LUT + AC. Blue arrows represent diffuse watery degeneration of hepatocytes; green arrows represent necrosis and swelling of the cells; yellow arrows show circular vacuoles of different sizes appearing in the cytoplasm.

**Figure 3 toxins-15-00478-f003:**
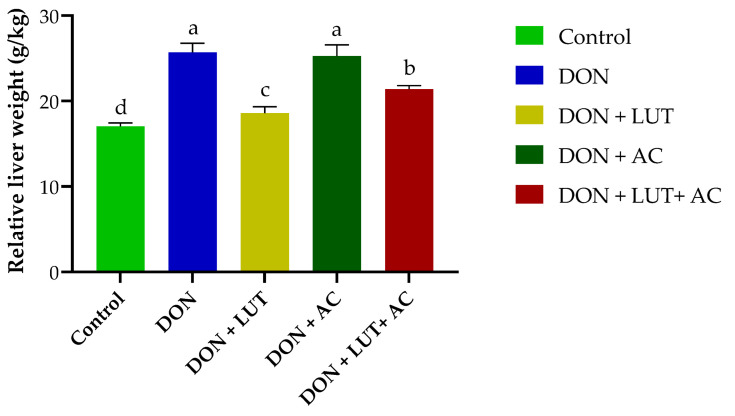
The effect of LUT and AC on DON-induced liver weight of broilers. Different letters a–d show significant differences.

**Figure 4 toxins-15-00478-f004:**
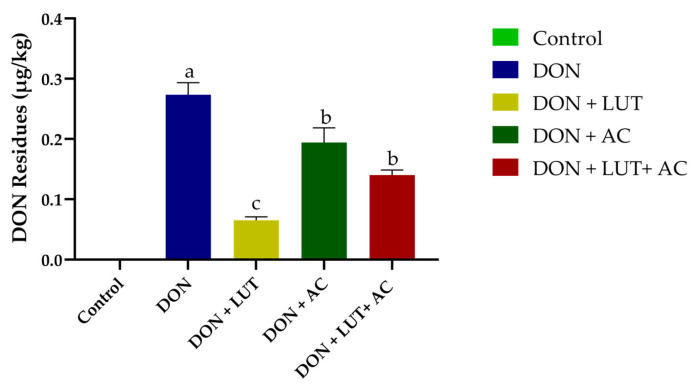
The effect of LUT and AC on DON residues in the liver of broilers. Significant differences were labeled with different letters a–c.

**Table 1 toxins-15-00478-t001:** Effect of LUT and AC on broiler growth performance.

Parameters	Control	DON	DON + LUT	DON + AC	DON + LUT + AC
ADFI (g/day)	67.73 ± 0.77 ^a^	47.77 ± 1.29 ^d^	63.90 ± 2.34 ^c^	65.68 ± 1.17 ^b^	64.86 ± 0.64 ^b c^
ADG (g/day)	34.59 ± 0.88 ^a^	25.46 ± 0.32 ^c^	30.31 ± 0.64 ^b^	33.74 ± 0.28 ^a^	31.57 ± 1.01 ^b^
FCR (feed: gain)	1.45 ± 0.012 ^d^	1.63 ± 0.014 ^a^	1.56 ± 0.07 ^b^	1.48 ± 0.012 ^d^	1.52 ± 0.04 ^c^

The data is statistically expressed as the mean ± SEM (n = 6). Significant differences were found in a row of mean values with different superscript letters ^a–d^ (*p* < 0.05). (DON: Deoxynivalenol, LUT: Luteolin, ADG: average daily gain, ADFI: average daily feed intake, and FCR: feed conversion ratio).

**Table 2 toxins-15-00478-t002:** Serum biochemical parameters of broilers fed DON-contaminated diets.

Parameters	Control	DON	DON + LUT	DON + AC	DON + LUT *+* AC
ALT (U/L)	1.74 ± 0.01 ^b^	2.85 ± 0.075 ^a^	1.68 ± 0.15 ^c^	1.53 ± 0.20 ^e^	1.57 ± 0.28 ^d^
AST (U/L)	197.89 ± 1.09 ^d^	265.64 ± 32.22 ^c^	173.60 ± 1.04 ^e^	223.78 ± 1.04 ^a^	213.68 ± 0.77 ^b^
ALP (U/L)	1346.63 ± 55.39 ^b^	1874.98 ± 80.98 ^a^	1329.78 ± 90.32 ^b c^	1237.51 ± 56.37 ^c^	1298.35 ± 64.05 ^b c^
TG (mmol/L)	0.93 ± 0.08 ^b^	1.37 ± 0.01 ^a^	0.87 ± 0.07 ^c^	0.52 ± 0.07 ^e^	0.76 ± 0.07 ^d^
TP (g/L)	33.42 ± 0.74 ^a^	20.77 ± 1.95 ^d^	32.52 ± 1.40 ^a^	27.92 ± 0.52 ^c^	30.51 ± 0.77 ^b^
Albumin (g/L)	17.94 ± 0.93 ^a^	14.83 ± 0.47 ^b^	17.84 ± 0.68 ^a^	15.83 ± 0.59 ^b^	17.15 ± 0.17 ^a^
Globulin (g/L)	12.79 ± 0.58 ^a^	8.49 ± 0.43 ^d^	11.76 ± 0.73 ^a b^	10.467 ± 0.65 ^c^	11.26 ± 0.84 ^b c^

The data are expressed as the mean ± SEM, and n = 6 is the number of chicks in each replicate. A row of mean values with different alphabet letters ^a–e^ showed significant differences (*p* < 0.05). (DON: deoxynivalenol, LUT: luteolin, AC: activated carbon, ALT: alanine aminotransferase, AST: aspartate aminotransferase, TG: triglyceride, ALP: alkaline phosphate, TP: total protein.

**Table 3 toxins-15-00478-t003:** Comparison of antioxidant enzymes in broiler serum fed a DON-contaminated diet with LUT and AC.

Parameters	Control	DON	DON + LUT	DON + AC	DON + LUT + AC
MDA (nmol/mL)	3.98 ± 0.24 ^b^	5.40 ± 0.52 ^a^	3.75 ± 0.23 ^bc^	3.25 ± 0.35 ^d^	3.57 ± 0.34 ^c^
T-SOD (U/mL)	142.11 ± 0.73 ^b^	105.58 ± 0.50 ^e^	140.67 ± 0.45 ^c^	132.28 ± 0.35 ^d^	148.43 ± 0.45 ^a^
GSH-Px (U/mL)	1420.05 ± 59.06 ^a^	894.37 ± 78.47 ^c^	1390.17 ± 125.35 ^a^	1586.67 ± 95.29 ^b^	1480.81 ± 85.48 ^b^
CAT (U/mL)	4.72 ± 0.18 ^a^	1.98 ± 0.19 ^c^	3.14 ± 0.21 ^a b^	5.96 ± 0.67 ^b^	4.02 ± 0.54 ^b^

Data showed as the mean ± SEM (n = 6). According to different alphabet letters ^a–e^, mean values differed significantly (*p* < 0.05). (GSH-Px: glutathione peroxidase, MDA: malondialdehyde, T-SOD: total superoxide dismutase, CAT: catalase).

**Table 4 toxins-15-00478-t004:** Broiler liver antioxidant parameters affected by LUT and AC in DON-contaminated diets.

Parameters	Control	DON	DON + LUT	DON + AC	DON + LUT *+* AC
CAT (U/mgprot)	82.76 ± 0.43 ^a^	38.73 ± 0.43 ^e^	78.63 ± 0.07 ^b^	69.43 ± 0.62 ^d^	75.83 ± 0.35 ^c^
GSH-Px (U/mgprot)	52.05 ± 0.28 ^a^	28.37 ± 0.330 ^e^	49.67 ± 0.47 ^b^	42.17 ± 0.51 ^d^	45.81 ± 0.27 ^c^
T-SOD (U/mgprot)	69.50 ± 0.33 ^a^	47.84 ± 0.51 ^e^	67.64 ± 0.47 ^b^	62.28 ± 0.56 ^d^	65.44 ± 0.31 ^c^
MDA (nmol/mgprot)	1.83 ± 0.02 ^b^	2.83 ± 0.29 ^a^	1.71 ± 0.06 ^b^	1.64 ± 0.35 ^b^	1.67 ± 0.38 ^b^

The data is represented in mean ± SEM (n = 6). Mean values in lines with dissimilar superscript letters ^a–e^ differed significantly (*p* < 0.05). CAT: catalase; MDA: malondialdehyde; GSH-Px: glutathione peroxidase, T-SOD: total superoxide dismutase.

**Table 5 toxins-15-00478-t005:** Formulation of a basal diet and evaluation of its nutritional value.

Ingredient	Percentage (%)
Corn	58.3
Soybean meal	30.2
Fish meal	5.6
Soybean oil	2.3
Dicalcium phosphate	1.2
Limestone	1.00
Salt	0.2
Methionine	0.2
Premix ^1^	1.00
Total	100
Calculated chemical composition	
Crude protein	21.87
Metabolizable energy (MJ/kg)	13.45
Lysine	1.14
Methionine	0.40
Methionine + Cystine	0.94
Calcium	0.95
Available phosphorus	0.49

^1^ The premix supplemented per kg of diet: Se, 0.4 mg; biotin, 0.04 mg; choline chloride, 400 mg; vitamin A (from retinyl acetate), 4500 IU; vitamin D3 (from cholecalciferol), 1000 IU; Fe, 60 mg; I, 1.1 mg; vitamin B11, 1 mg; vitamin B12, 0.013 mg; vitamin K (menadione sodium bisulfate), 1.3 mg; vitamin B1, 2.2 mg; vitamin B2, 10 mg; vitamin B3, 10 mg; vitamin B5, 50 mg; vitamin B6, 4 mg; Cu, 7.5 mg; Zn, 65 mg; Mn, 110 mg.

## Data Availability

The data presented in this study are available on request from the corresponding author.

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
