# Peer review of "Ameliorative Effects of Luteolin and Activated Charcoal on Growth Performance, Immunity Function, and Antioxidant Capacity in Broiler Chickens Exposed to Deoxynivalenol"

_toxins, 2023, doi:10.3390/toxins15080478_

Round 1

Reviewer 1 Report

The authors describe a study on the effectiveness of luteolin in combination with activated charcoal in detoxifying DON in broilers. Several studies in the literature have searched for new possibilities for the physical, chemical, and biological detoxification of DON. This study should represent a good addition to the literature showing promising results of dealing with DON using physical-chemical detoxification. The results reached in this study are relevant and the experiments have been well carried out. For the above reasons, it can be accepted for publication in Toxins after a minor revision.

Note: No extensive grammar corrections were performed in the text. Please, assure that the proper language used in all text is corrected and extensively verified by professional English editing services before being taken for publication.

Please verify the following:

Title

The title is too long, please adjust this.

Materials and methods

Explain the origin or how it was acquired the luteolin and activated charcoal used in the study.

It is known that the contact time of the mycotoxin with the adsorbent can influence its efficiency. Inform if some reference was used for determine the time of contact of the mycotoxin with the adsorbent used for the experiment.

Explain better the detection of DON in the HPLC with a fluorescence detector. Please, inform if some derivatizing was used in the procedure and added the reference.

Results

Page 2, line 85

The following information is not clear. “Additionally, the LUT and AC treatment did not negatively affect broiler performance.” It seems that treatment only with LUT and AC (without DON) was carried out.

Page 3, lines 107-109

Add the ALB, albumin.

Figure 3

There doesn't seem to be a significant difference between DON and DON + AC, please to check it.

Page 5, line 170

Adjust, instead of “letters a-d” replace “letter a-e”.

Figure 4

It seems that LUT has some effect on the DON molecule since the DON + LUT treatment reduced DON residues. This should be explained in the discussion.

Conclusion

Add what are the prospects for further studies. Inform if LUT and AC have the potential to prevent damage caused by other mycotoxins.

Reviewer 2 Report

The manuscript describes a study investigating the effect of luteolin and activated charcoal on deoxynivalenol (DON)-exposed broiler chicken. The manuscript is well written but lacks some information in the material and methods second. Furthermore, I have some doubt in the determination of the mycotoxin content since it was done by an ELISA test kit which does not take into account any other mycotoxin except DON.

Line 13: Please specify the number of replicates “sie” is not a number.

Line 14: “Ad libitum should be written in italics.

Line 14: It should be specified that the basal diet did not contain LUT, DON or AC (not just “not LUT and DON).

Line 15ff: Why do you always use a capital starting letter after the “;” – it should always be a small letter.

Line 19: I would not call the “control” group” “placebo group” – it is not a medication.

Line 23: Please explain the used abbreviation “SOD”. All abbreviations should be introduced the first time used.

Keywords: Please check with the editorial office. Usually in the keywords abbreviations should be avoided.

Line 35: Please consider rephrasing since the “E.U.” is not a country, but a union of several countries.

Line 70: “in vitro” should be written in italics.

Line 77: Please use the correct abbreviations “DON” and “LUT”

Table 2: Please think about the significant number of digits you can report. Six significant numbers as in the case of ALP seem a little bit too much. In my opinion, three or a maximum of four significant numbers are enough (also concerns Table 1 to a lesser extend).

The brackets around the unit of TG “(mmol/L)” are missing.

Figure 1: Since in the manuscript text itself and in the Tables you used the expression “mL” you should also replace the y-axis “mg/ml” with “mg/mL” in this Figure.

Does it make sense to compare the levels of different serum immunoglobulins directly? I would only compare the different groups of one Ig. Then the DON bar of IgM should be labelled with “c” instead of “d”.

Table 3: Following the style in Table 2, the units should be always displayed in brackets (concerns also Table 4). Also in this Table, please think about the significant number of digits you can report.

Line 138: Please check whether or not “Fig. 2A-C” is correct since B shows the DON-group. Furthermore, the verb is missing in the second part of the sentence “while little changes in the DON+LU+AC groups WERE OBSERVE”.

I am not a histopathology specialist and for me it is difficult to assess the different pictures. What are the white areas displayed in A, C and D? The morphology of E seems to be quite different compared to the others in this picture – there is far more “white” areas around the “pink islands”.

Line 169: Please remove the second “significant” since it does not make sense to state it twice in this context. In which order do you provide the abbreviations? It is not according to the alphabet and not according to the order in the Table (also concerns the other Tables).

Line 172ff: Did you only look for DON or also for DON-metabolites in the liver? Might be worth, to look for sulfate conjugates or conjugation with glucuronide.

Line 175: Why do you state that DON+LUT did not exhibit DON residues in the liver, but in Figure 4, there is a bar until 0.1 µg/kg.

Figure 4: Please check the unit on the y-axis. Do you mean “µg/kg”?

Discussion:

When you discuss your results, it might be good to refer to the respective table to underline the statement. E.g., in line 223 you could refer to Table 1.

Line 189: It should be “that 15 mg/kg DON reduced broiler …”.

Line 190: Please check, but in my opinion it should be “after 42 days”.

Line 208: Please indicate the genus name of “S.” (Saccharomyces) and write the genus and species name in italics.

Line 218: Please explain the abbreviation (Zen) which should be written with all capital letters.

Line 222: Please explain the used abbreviation (AFB1).

Line 245, 255, 260, 271: Please be consistent and use always the abbreviation “DON” instead of “vomitoxin” in this context.

Material and Methods:

I miss a little bit more detail on the applied methods. The way it is described it is not easily reproducible. Please consider to provide more detail.

Line 291: The “6” in “1x106” should be superscripted.

Line 294: Why did you not use the later on described HPLC-method for this purpose? The ELISA Kit is not validated for fungal cultures since the concentration might be too high – did you dilute the samples? Did you also check for the production of 3-acetyl-deoxynivalenol or 15-acetyl-deoxynivalenol – the fungal precursors of DON? Furthermore, did you check whether this specific strain is also producing zearaleneone (ZEN) – a mycotoxin often co-produced by Fusarium graminearum. How did you make sure that the fungus was not growing further on the contaminated died increasing the DON load?

Line 296: The “®” should be superscripted.

Line 297: The result of this absorption study – are they provided anywhere? I could not find it.

Line 298: The word “either” is usually used in the context “either x or y”, so I am missing the “or y” in this context.

Line 312: Now I am a little bit confused. I thought that you used six animals per group, so why do you write “five groups of three repetitions each” and then write “(n=6)”. Where there three or six animals in each group?

Line 314: I am not sure why you wrote here the full name “Basal diet”, when you abbreviated it in all other groups.

Line 319: Is it normal to expose the chicks the whole day and then have only one hour of darkness? I have never heard of it before. Thought that a very similar day/night cycle is followed as natural exposure.

How many chicks were in one battery?

Line 329: You specify the absence of several mycotoxins, but what about DON?

Line 362ff> I am not 100% sure whether I understood it correctly. So, you analysed only one bird per group – this would be five birds for five groups, why did you have six birds? However, how can you then display a standard deviation? Please check and rephrase if necessary.

Line 365: You filtered the samples, but you did not add any solvent? Please specify and provide a little bit more details on the method – the polarity of aflatoxins and DON are not the same. So did you validate the method for this purpose? How did you derivatize the solution?

References

Please make sure that the species names are written in italics (e.g., line 416 Coturnix Japonica, line 436 Manihot Esculenta Crantz, line 443 Campylobacter jejuni, line 495 Staphylococcus aureus, line 508 Aspergillus flavus, line 511 Aspergillus parasiticus). Usually also the species name is written with a small starting letter in contrast to the genus name which is written with a capital starting letter.

Furthermore, “in vitro”, “in vivo” should be written in italics (e.g., line 455 and 457).

Please check reference 16: Usually, names are written with capital starting letters.

Line 462: The usual convention is that aflatoxin B1 is written with a capital “B”.

You might want to include the following recent publication of the European Commission:
https://www.efsa.europa.eu/en/efsajournal/pub/7806

The quality of the manuscript with regard to the English language is fine - only minor adaptations are necessary.

Round 2

Reviewer 2 Report

Thank you very much for revising the manuscript. The quality improved a lot, but I still have a few comments prior to publication.

Title: I suggest to move “Ca-“ into the next line since in my opinion it is not advantageous that such a short word (Capacity) is displayed in two lines.

Line 15: Here a “+” between the concentration of DON and AC is missing. Please add.

Key Contributions: I would add the time frame of the investigation since it is a very important information.

Introduction:

Line 39: Please specify the timeframe of the increase of poultry consumption.

Line 48: You state “barley” twice, please delete it once.

Results & Discussion:

Line 146: You state here significances of “a-d”, but in the figure there are only “a-c” please check.

Line 153: What does “H & E” stand for? Please provide the full name of the abbreviation.

Figure 3: Please check the displayed significance values. I think the column DON + AC should have the letter “a” and the column DON + LUT + AC the letter “b”, DON+LUT letter “c” and control “letter “d”.

Line 170ff: Please think about the significant number of digits you use in this context for reporting the percentages. In my opinion two or a maximum of three are enough (instead of four). This also concerns line 184 and the remaining manuscript.

Line 227 and Line 230: The names of the mycotoxins (zearalenone and aflatoxin B1) should be written with small starting letters.

Line 295: In my opinion it should be “duck livers” (not duck`s livers).

Materials and Methods:

Line 322: Which HPLC method did you use for the assessment?

Line 385: The approach described in reference 54 is an LC-MS/MS approach using stable isotopically labelled standards. I do not have access to reference 56 (and it might be the same for other readers). Therefore, please state at least the derivatisation procedure you applied since DON is not fluorescent per se.

In large parts of the manuscript the quality of English is ok, but some improvements are necessary.
